# The chromatin-remodeling enzyme CHD3 plays a role in embryonic viability but is dispensable for early vascular development

Jun Xie[1], Siqi Gao[1,2], Christopher Schafer[1], Sarah Colijn[1,2¤a], Vijay Muthukumar[1¤b], Courtney T. Griffin[1,2]*

**1** Cardiovascular Biology Research Program, Oklahoma Medical Research Foundation, Oklahoma City, OK, United States of America, **2** Department of Cell Biology, University of Oklahoma Health Sciences Center, Oklahoma City, OK, United States of America

¤a Current address: Department of Cell Biology & Physiology, Washington University School of Medicine, St. Louis, MO, United States of America
¤b Current address: The Jackson Laboratory, Bar Harbor, ME, United States of America
* courtney-griffin@omrf.org

**Data Availability Statement:** All relevant data are within the paper.

## Abstract

ATP-dependent chromatin-remodeling complexes epigenetically modulate transcription of target genes to impact a variety of developmental processes. Our lab previously demonstrated that CHD4—a central ATPase and catalytic enzyme of the NuRD chromatin-remodeling complex—plays an important role in murine embryonic endothelial cells by transcriptionally regulating vascular integrity at midgestation. Since NuRD complexes can incorporate the ATPase CHD3 as an alternative to CHD4, we questioned whether the CHD3 enzyme likewise modulates vascular development or integrity. We generated a floxed allele of *Chd3* but saw no evidence of lethality or vascular anomalies when we deleted it in embryonic endothelial cells in vivo (*Chd3^ECKO*). Furthermore, double-deletion of *Chd3* and *Chd4* in embryonic endothelial cells (*Chd3/4^ECKO*) did not dramatically alter the timing and severity of embryonic phenotypes seen in *Chd4^ECKO* mutants, indicating that CHD3 does not play a cooperative role with CHD4 in early vascular development. However, excision of *Chd3* at the epiblast stage of development with a *Sox2-Cre* line allowed us to generate global heterozygous *Chd3* mice (*Chd3^Δ/+*), which were subsequently intercrossed and revealed partial lethality of *Chd3^Δ/Δ* mutants prior to weaning. Tissues from surviving *Chd3^Δ/Δ* mutants helped us confirm that CHD3 was efficiently deleted in these animals and that CHD3 is highly expressed in the gonads and brains of adult wildtype mice. Therefore, *Chd3*-flox mice will be beneficial for future studies about roles for this chromatin-remodeling enzyme in viable embryonic development and in gonadal and brain physiology.

## Introduction

The mammalian <u>N</u>ucleosome <u>R</u>emodeling and Histone <u>D</u>eacetylase (NuRD/Mi-2) complex is an ATP-dependent chromatin-remodeling complex that can transiently displace nucleosomes

**Funding:** This work was supported by grants from the National Institutes of Health (https://www.nih.gov) awarded to C.T.G. (R35HL144605) and to Rodger McEver (P30GM114731). The funders had no role in study design, data collection and analysis, decision to publish, or preparation of the manuscript.

**Competing interests:** The authors have declared that no competing interests exist.

within gene regulatory regions and thereby impact transcription [1]. NuRD complexes are composed of multiple proteins, many of which have varying isoforms that can contribute to differential NuRD assembly in specific cell types and tissues [2, 3]. Among these proteins, the Chromodomain-Helicase-DNA-Binding ATPases CHD3 and CHD4 (also called Mi-2α and Mi-2β) are critical for NuRD activity because they provide the energy and helicase function required for chromatin remodeling [4]. Recent evidence suggests that CHD3 and CHD4 are mutually exclusive within NuRD complexes, despite the fact that they are co-expressed in various cell types [5, 6].

NuRD complexes play various roles in developmental and cellular differentiation processes, such as suppression of embryonic stem cell pluripotency, maintenance and lineage commitment of hematopoietic stem cells, maintenance of nephron progenitors in the kidney, peripheral nerve myelination, and cortical development in the brain [6–10]. Many of these and additional discoveries resulted from studies focused on genetic deletion of a single NuRD component or a NuRD-interacting cofactor [11]. A *Chd4*-flox allele generated by Katia Georgopoulos' lab [12] has been particularly useful for defining the function of CHD4-NuRD complexes in specific cell types in vivo.

Our lab has a long-standing interest in the role of chromatin-remodeling enzymes in vascular development. We genetically delete chromatin-remodeling enzymes like CHD4 in murine embryonic endothelial cells in order to determine the impact on developing blood vessels and to identify target genes that require tight transcriptional regulation during vascular development [13]. We previously reported that CHD4 is required for maintenance of vascular integrity at midgestation and during liver development [14–16]. We also described how CHD4 counteracts the activity of the chromatin-remodeling enzyme BRG1 to regulate the Wnt signaling pathway in developing blood vessels [17]. Together these studies demonstrate that CHD4 plays critical roles in embryonic vascular development and maintenance, presumably through its participation in NuRD-related transcriptional regulation events in endothelial cells.

Because of the interesting and varied vascular phenotypes we saw associated with *Chd4* deletion in developing vasculature, we wondered what role the alternative NuRD ATPase CHD3 might play in embryonic blood vessels. No knockout or conditional allele of *Chd3* currently exists for genetic deletion studies in mice. Instead, one report describes electroporating shRNAs against CHD3 into embryonic mouse brains in order to inhibit CHD3 in vivo [6]. In this study we sought to analyze CHD3 during vascular development by generating a *Chd3*-flox allele and deleting the gene in developing endothelial cells. Although we found no obvious roles for CHD3 in early vascular development, global deletion of the gene resulted in partial embryonic lethality. Our genetic crosses and expression data from adult mouse tissues—along with recent reports about CHD3 roles in neural development—suggest that CHD3 may contribute to viable embryonic brain development.

## Materials and methods

### Mice

All animal use protocols were approved by the Oklahoma Medical Research Foundation Institutional Animal Care and Use Committee (protocol #20–15), and all efforts were made to minimize animal suffering. Animals were originally described and obtained from the following sources: *Chd3*-flox, generated by Cyagen Biosciences as described in this paper; *Chd4*-flox [12], gift of Dr. Katia Georgopoulos (Harvard Medical School); *Tie2-Cre* [18], The Jackson Laboratory #008863; *Sox2-Cre* [19], The Jackson Laboratory #008454; and FLPe [20], The Jackson Laboratory #005703. Timed matings were assessed by checking daily for a copulation plug;

noon on the day of plug detection was designated as embryonic day 0.5 (E0.5). Embryos and yolk sacs were dissected from maternal tissue, and yolk sacs were digested for genotyping assays. Gross embryonic images were obtained with a Nikon SMZ800 stereomicroscope and Nikon DS-Fi1 camera and monitor. Genotyping primers and conditions are described in Table 1.

## Immunoblotting

Total protein harvested from eight-week old mouse tissues was quantified with the Pierce BCA Protein Assay Kit (Thermo Scientific). Protein (20–30μg) was fractionated on a 9% SDS-poly-acrylamide gel and transferred to a polyvinylidene difluoride (PVDF) membrane for immuno-blotting with one of two antibodies against CHD3 (1:2000; BD Biosciences; Cat. #611847; RRID: AB_399327; or 1:1000; Abcam; Cat. #ab109195; RRID: AB_10862514) and GAPDH (1:5000; Sigma; Cat. #9545; RRID: AB_796208). Horseradish peroxidase-conjugated secondary antibodies and SuperSignal West Femto Maximum Sensitivity Substrate (Thermo Scientific) were used for detection. Imaging was performed on a Fluorchem HD2 (Alpha Innotech) and scanned with an imageRUNNER 3225 (Canon).

## Statistics

Statistical methodology is described in the table legends.

## Results

### Generation of a *Chd3*-flox allele

In order to generate a *Chd3*-flox allele to facilitate conditional deletion of the chromatin-remodeling enzyme in vivo, we contracted with Cyagen Biosciences to design a targeting vector for homologous recombination in embryonic stem (ES) cells (Fig 1A–1D). We chose to flank exons 13 through 20 of the murine *Chd3* gene with loxP sites, since the CHD3 ATPase helicase—which mediates the critical function of the protein [21]—is encoded in this region. Deletion of these exons via Cre-mediated recombination was predicted to result in loss of function of the *Chd3* gene by generating a frameshift mutation. Homologous recombination of the targeting vector in ES cells was confirmed by Cyagen Biosciences by Southern blotting

**Table 1. Genotyping primers and conditions.**

| Allele | Genotyping Primers | | Annealing Temperature | Amplicon Size |
|---|---|---|---|---|
| *Chd3*-flox | Forward (F): 5′-GGGTGGAGGTGGAAAGTGTA-3′ | | 55˚C | Wildtype: 152 bp |
| | Reverse (R1): 5′-AGAGGACAGGTCACAGGACAA-3′ | | | Floxed: 213 bp |
| *Chd3*$^\Delta$ | Forward (F): 5′-GGGTGGAGGTGGAAAGTGTA-3′ | | 59˚C | ~230 bp |
| | Reverse (R2): 5′-GGCACAGCTTCTCTGAGACC-3′ | | | |
| *Chd4*-flox | Forward: 5′-TCCAGAAGAAGACGGCAGAT-3′ | | 55˚C | Wildtype: 278 bp |
| | Reverse: 5′- CTGGTCATAGGGCAGGTCTC-3′ | | | Floxed: 400 bp |
| *Chd4*$^\Delta$ | Forward: 5′-TCCAGAAGAAGACGGCAGAT-3′ | | 59˚C | ~350 bp |
| | Reverse: 5′-CAGCGTGTCAGAGTTTGCAT-3′ | | | |
| *Tie2-Cre* | Forward: 5′-GGGAAGTCGCAAAGTTGTGAGTTG-3′ | | 60˚C | 533 bp |
| | Reverse: 5′-TCCATGAGTGAACGAACCTGGTCG-3′ | | | |
| *Sox2-Cre* | Forward: 5′-TGCAACGAGTGATGAGGTTC-3′ | | 58˚C | 528 bp |
| | Reverse: 5′-ATTCTCCCACCGTCAGTACG-3′ | | | |
| *FLPe* | Forward: 5′-AGTAGTGATCAGGTATTGCTGTTATCTG-3′ | | 60˚C | 100 bp |
| | Reverse: 5′-GACTAATGTTGTGGGAAATTGGAG-3′ | | | |

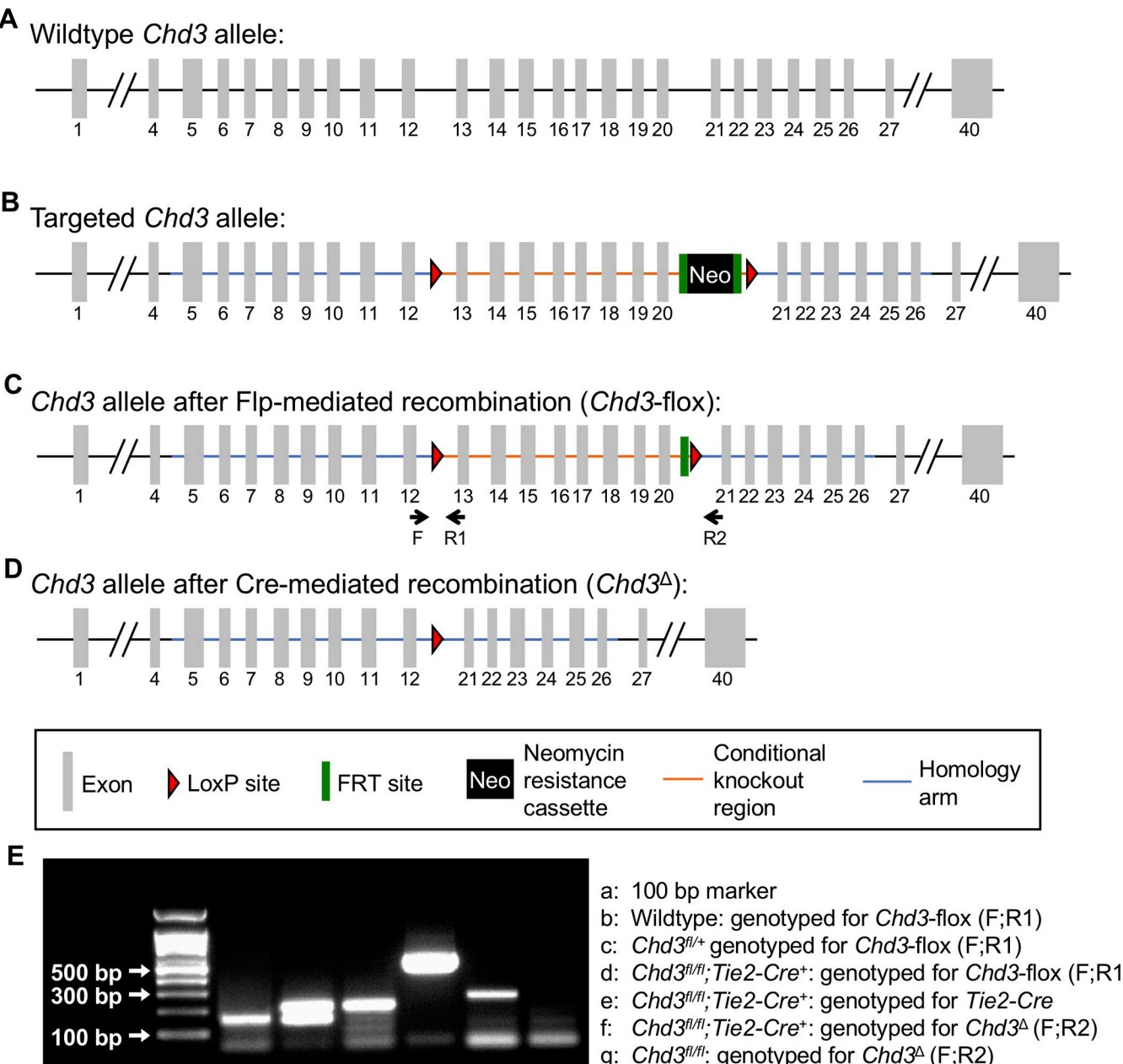

**Fig 1. Generation of a conditional murine *Chd3* knockout allele.** The *Chd3*-flox allele was generated using homologous recombination of a targeting vector in C57Bl/6 embryonic stem cells. (A): The wildtype murine *Chd3* allele contains 40 exons and resides on chromosome 11. (B): The targeting vector incorporated a 5' LoxP site (red triangle) inserted between exons 12 and 13 and a 3' LoxP site inserted between exons 20 and 21 of the wildtype *Chd3* allele. In addition, a neomycin resistance (Neo) cassette flanked by Frt sites (green bars) was inserted immediately upstream of the 3' LoxP site for positive selection of the integrated targeting vector. (C): The Neo cassette was removed by crossing mice carrying the targeted allele to FLPe-recombinase mice, leaving a single Frt site upstream of the 3' LoxP site in the *Chd3-flox* allele after recombination. Genotyping primers are indicated below the allele. (D): After crossing mice carrying the *Chd3-flox* allele to a line carrying a Cre recombinase, exons 13–20 were deleted (*Chd3$^\Delta$*). (E): The indicated mice were genotyped by PCR using the primers shown in parentheses and diagrammed in (C) to demonstrate targeting and Cre-mediated excision of the *Chd3*-flox allele. PCR products were run on a 2% agarose gel. See Table 1 for primer sequences and predicted amplicon sizes.

prior to blastocyst injection. Targeting and excision events were confirmed by PCR in F1 and subsequent generations of mice (Fig 1E).

## CHD3 is not essential for embryonic vascular development or for adult vascular maintenance

In order to determine whether CHD3 plays a critical role in vascular development or maintenance, we next crossed the *Chd3*-flox allele onto a transgenic *Tie2-Cre* recombinase line that is expressed in embryonic endothelial and hematopoietic cells [18]. We found that *Chd3*$^{fl/fl}$; *Tie2-Cre*$^+$ mice (hereafter, *Chd3*$^{ECKO}$ for *Chd3*-endothelial cell knockout) were born in expected numbers and survived past weaning and into adulthood without obvious abnormalities (Table 2).

Although our genetic crosses indicated that no lethal vascular or hematopoietic abnormalities arose from *Chd3* deletion with the *Tie2-Cre* recombinase, we visually analyzed *Chd3*$^{ECKO}$ embryos at midgestation, since that is the developmental stage at which we had previously observed significant vascular phenotypes in *Chd4*$^{fl/fl}$;*Tie2-Cre*$^+$ (hereafter *Chd4*$^{ECKO}$) embryos [14, 15]. Specifically, we dissected *Chd3*$^{ECKO}$ and *Chd4*$^{ECKO}$ embryos at E11.5; while the *Chd4*$^{ECKO}$ embryos displayed lethal vascular rupture and hemorrhage, as we previously described, the *Chd3*$^{ECKO}$ embryos were indistinguishable from control littermates (Fig 2A–2C).

## Vascular phenotypes in *Chd3*$^{fl/fl}$;*Chd4*$^{fl/fl}$;*Tie2-Cre*$^+$ (*Chd3/4*$^{ECKO}$) embryos resemble those seen in *Chd4*$^{ECKO}$ mutants

Because we have reported that chromatin-remodeling enzymes can co-regulate the same target genes or signaling pathways in developing blood vessels [17], we next sought to determine whether CHD3 plays a cooperative or antagonistic role with CHD4 during embryonic vascular development. Specifically, we generated embryos deficient for both endothelial *Chd3* and *Chd4* (*Chd3*$^{fl/fl}$;*Chd4*$^{fl/fl}$;*Tie2-Cre*$^+$, hereafter *Chd3/4*$^{ECKO}$) to determine whether the vascular phenotype was different from that seen in littermate *Chd4*$^{ECKO}$ embryos. We observed a comparable onset of lethality in *Chd3/4*$^{ECKO}$ and *Chd4*$^{ECKO}$ embryos between E10.5–11.5 (Table 3 and Fig 2D). Therefore, endothelial and hematopoietic cell deletion of *Chd3* did not rescue lethal *Chd4*$^{ECKO}$ vascular phenotypes at midgestation. In addition, we assessed *Chd3/4*$^{ECKO}$ embryos at E10.5—one day prior to the vascular rupture seen in *Chd4*$^{ECKO}$ embryos—and saw no indication of vascular abnormalities (Fig 3). Therefore, *Tie2-Cre*-mediated deletion of *Chd3* did not exacerbate the timing or severity of *Chd4*$^{ECKO}$ vascular phenotypes. Altogether, these genetic data indicate that CHD4 plays a more consequential role than does CHD3 in NuRD-mediated chromatin-remodeling activities during early vascular development.

**Table 2. Live offspring at weaning from *Chd3*$^{fl/+}$ X *Chd3*$^{fl/+}$;*Tie2-Cre*$^+$ crosses**[a].

| Genotype | Observed Offspring[b] | Expected Offspring |
|---|---|---|
| *Chd3*$^{+/+}$ | 6 | 9.5 |
| *Chd3*$^{fl/+}$ | 29 | 19 |
| *Chd3*$^{fl/fl}$ | 8 | 9.5 |
| *Chd3*$^{+/+}$;*Tie2-Cre*$^+$ | 9 | 9.5 |
| *Chd3*$^{fl/+}$;*Tie2-Cre*$^+$ | 15 | 19 |
| *Chd3*$^{fl/fl}$;*Tie2-Cre*$^+$ (*Chd3*$^{ECKO}$) | 9 | 9.5 |

***Chd3*$^{fl/fl}$;*Tie2-Cre*$^+$ (*Chd3*$^{ECKO}$) mice survive development.** 76 mouse pups generated from matings between *Chd3*$^{fl/+}$;*Tie2-Cre*$^+$ males and *Chd3*$^{fl/+}$ females were genotyped at weaning (~19 days). No lethal embryonic vascular anomalies are associated with the *Chd3*$^{ECKO}$ genotype, since the numbers of *Chd3*$^{fl/fl}$;*Tie2-Cre*$^+$ progeny observed approximate those expected.

[a]11 litters generated from 8 *Chd3*$^{fl/+}$ females and 5 *Chd3*$^{fl/+}$;*Tie2-Cre*$^+$ males were genotyped.

[b]$\chi^2$(5$_{dof}$) = 7.68, 0.5>P>0.1.

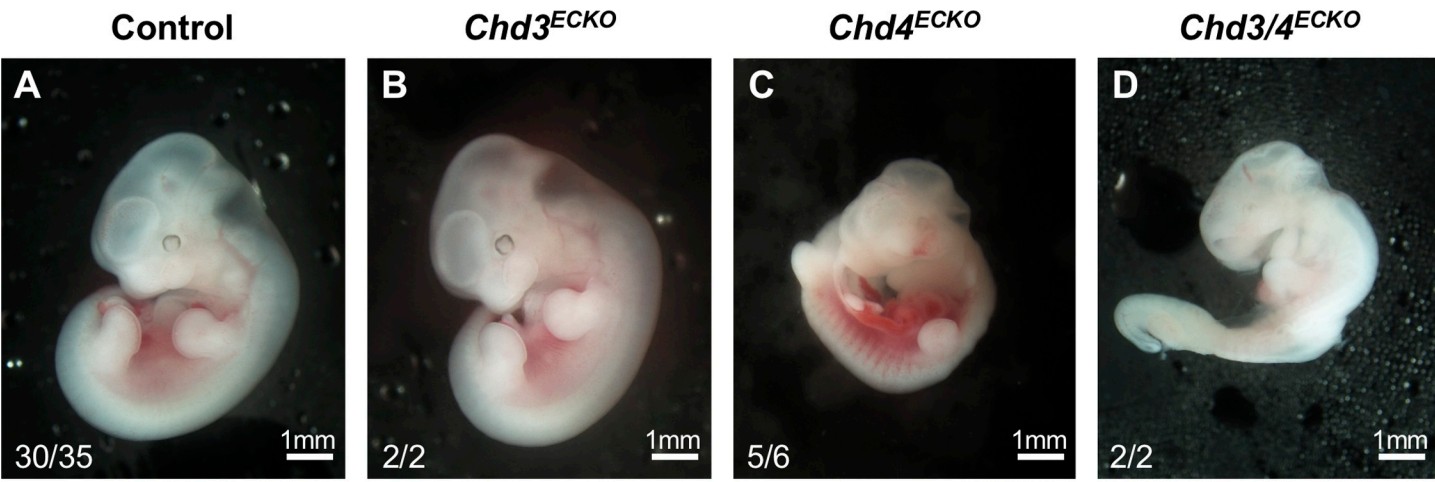

**Fig 2. *Chd3^ECKO* embryos are alive and display normal gross morphology at E11.5.** Photographs of representative E11.5 littermate embryos are shown. Each embryo was pronounced alive or dead at the time of dissection based on the presence or absence of a visible heartbeat. Numbers in the bottom left corner of each picture report the embryos resembling that picture/total embryos of each genotype dissected at E11.5. (A): Control = *Tie2-Cre^-* (B): *Chd3^ECKO* = *Chd3^fl/fl*;*Tie2-Cre^+* (C): *Chd4^ECKO* = *Chd4^fl/fl*; *Tie2-Cre^+* (D): *Chd3/4^ECKO* = *Chd3^fl/fl*;*Chd4^fl/fl*;*Tie2-Cre^+*.

## Global deletion of *Chd3* results in partial embryonic lethality

Since we saw no visible phenotypes associated with deletion of *Chd3* in blood vessel endothelial cells and hematopoietic cells, we next sought to determine the consequence of global excision of our *Chd3*-flox allele. We crossed the *Chd3*-flox line onto a *Sox2-Cre^+* recombinase line that is expressed in all embryonic cells starting at the epiblast stage of development [19]. We found an underrepresentation of *Chd3^fl/fl*;*Sox2-Cre^+* (hereafter called *Chd3^Δ/Δ*;*Sox2-Cre^+* or *Chd3*-knockout) mice at weaning (Table 4). Likewise, matings between *Chd3^Δ/fl* (*Chd3*-heterozygous) mice confirmed partial lethality of *Chd3^Δ/Δ* (*Chd3*-knockout) mice (Table 5). Because we did not observe noticeable lethality of pups between birth and weaning when performing these crosses, we believe that *Chd3*-knockout embryos died before birth.

**Table 3. Total numbers of embryos genotyped (and % dead) at midgestation.**

| Genotype | E10.5 | E11.5 |
|---|---|---|
| Control | 74 (3%) | 35 (14%) |
| *Chd3^ECHet* and/or *Chd4^ECHet* | 40 (1%) | 19 (0%) |
| *Chd3^ECKO* | 26 (12%) | 2 (0%) |
| *Chd4^ECKO* | 32 (6%) | 6 (83%) |
| *Chd3/4^ECKO* | 8 (12%) | 2 (100%) |

**The majority of *Chd3/4^ECKO* embryos die between E10.5–11.5.** Embryos were dissected at E10.5 or E11.5 from various matings that produce the following genotypes: Control (*Tie2-Cre^-*), *Chd3^ECHet* (*Chd3^fl/+*;*Tie2-Cre^+*), *Chd4^ECHet* (*Chd4^fl/+*;*Tie2-Cre^+*), *Chd3^ECKO* (*Chd3^fl/fl*;*Tie2-Cre^+*), *Chd4^ECKO* (*Chd4^fl/fl*;*Tie2-Cre^+*), or *Chd3/4^ECKO* (*Chd3^fl/fl*;*Chd4^fl/fl*;*Tie2-Cre^+*). All embryos were deemed "alive" or "dead" at the time of dissection based on the presence or absence of a visible heartbeat. The death of most *Chd4^ECKO* embryos between E10.5–11.5 is consistent with our previous reports [14, 15].

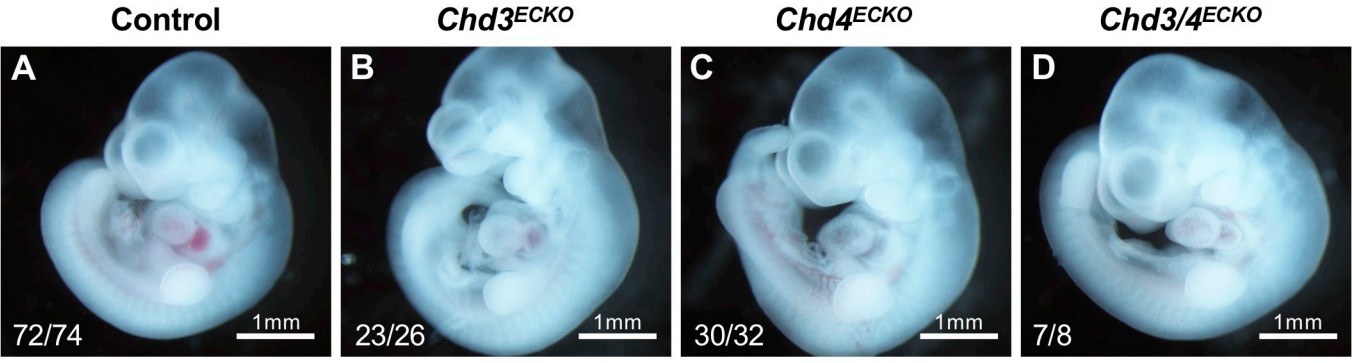

**Fig 3. *Chd3/4^ECKO^* embryos display normal gross morphology at E10.5.** Photographs of representative live E10.5 littermate embryos are shown. Numbers in the bottom left corner of each picture report live embryos/total embryos of each genotype dissected at E10.5. (A): Control = *Tie2-Cre*⁻ (B): *Chd3^ECKO^* = *Chd3^fl/fl^*; *Tie2-Cre*⁺ (C): *Chd4^ECKO^* = *Chd4^fl/fl^*;*Tie2-Cre*⁺ (D): *Chd3/4^ECKO^* = *Chd3^fl/fl^*;*Chd4^fl/fl^*;*Tie2-Cre*⁺.

## CHD3 is predominantly expressed in adult gonads and brain tissue

We next surveyed CHD3 expression in adult tissues, with hopes of gaining insight into possible causes of lethality for *Chd3^Δ/Δ^* animals. We harvested a variety of tissues from wildtype male and female mice and analyzed CHD3 expression by immunoblotting. We detected CHD3 protein predominantly in the brains (of both sexes), testes, and ovaries of eight-week-old mice (Fig 4). In order to validate the CHD3 antibodies we utilized for immunoblotting and to confirm the efficiency of *Chd3* deletion with the *Sox2-Cre* recombinase, we assessed CHD3 expression in adult littermate wildtype, *Chd3*-heterozygous (*Chd3^Δ/fl^*), and surviving *Chd3*-knockout (*Chd3^Δ/Δ^*) mice (Fig 5). Immunoblotting for CHD3 protein in brain tissue revealed the expected reduction of CHD3 expression in *Chd3*-heterozygous samples and absence of detectable CHD3 in *Chd3*-knockout samples. Therefore, our *Chd3*-flox allele can be excised efficiently. Moreover, our expression analyses suggest that the partial pre-weaning lethality we observe on the *Chd3*-knockout background could be due to compromised fertility or to fatal brain defects, such as the cortical abnormalities recently described when CHD3 is knocked down in developing mouse brains [6].

**Table 4. Live offspring at weaning from *Chd3^fl/+^*;*Sox2-Cre*⁺ (*Chd3^Δ/+^*;*Sox2-Cre*⁺) X *Chd3^fl/+^* crosses[a].**

| Genotype | Observed Offspring[b] | Expected Offspring |
|---|---|---|
| *Chd3^+/+^* | 21 | 25.375 |
| *Chd3^fl/+^* | 22 | 25.375 |
| *Chd3^Δ/+^* | 26 | 25.375 |
| *Chd3^Δ/fl^* | 35 | 25.375 |
| *Chd3^+/+^*;*Sox2-Cre*⁺ | 33 | 25.375 |
| *Chd3^Δ/+^*;*Sox2-Cre*⁺ | 53 | 50.75 |
| *Chd3^Δ/Δ^*;*Sox2-Cre*⁺ | 13 | 25.375 |

***Chd3^Δ/Δ^*;*Sox2-Cre*⁺ progeny are underrepresented in offspring from *Chd3^Δ/+^*;*Sox2-Cre*⁺ males and *Chd3^fl/+^* females.** 203 mouse pups generated from matings between *Chd3^Δ/+^*;*Sox2-Cre*⁺ males and *Chd3^fl/+^* females were genotyped at weaning (~19 days).

[a] 19 litters generated from 5 *Chd3^fl/+^* females and 4 *Chd3^Δ/+^*;*Sox2-Cre*⁺ males were genotyped.

[b] $\chi^2(6_{dof}) = 13.30$, $P<0.05$.

**Table 5. Live offspring at weaning from *Chd3^{Δ/fl}* X *Chd3^{Δ/fl}* crosses[a].**

| Genotype | Observed Offspring[b] | Expected Offspring |
|---|---|---|
| *Chd3^{fl/fl}* | 19 | 15 |
| *Chd3^{Δ/fl}* | 35 | 30 |
| *Chd3^{Δ/Δ}* | 6 | 15 |

***Chd3^{Δ/Δ}* progeny are underrepresented in offspring from *Chd3^{Δ/fl}* matings.** 60 mouse pups generated from matings between *Chd3^{Δ/fl}* males and females were genotyped at weaning (~19 days).

[a]7 litters generated from 2 *Chd3^{Δ/fl}* males and 2 *Chd3^{Δ/fl}* females were genotyped.

[b]$\chi^2(2_{dof}) = 7.3$, $P<0.05$.

## A    Male Tissues

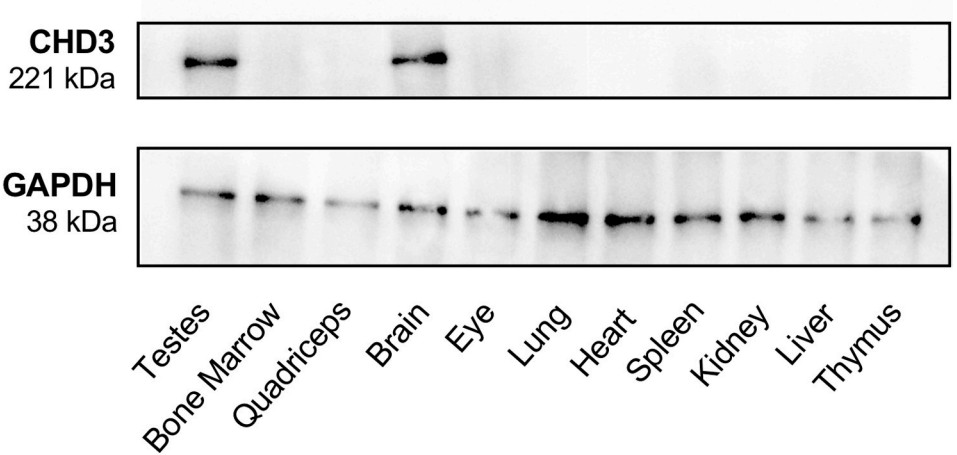

## B    Female Tissues

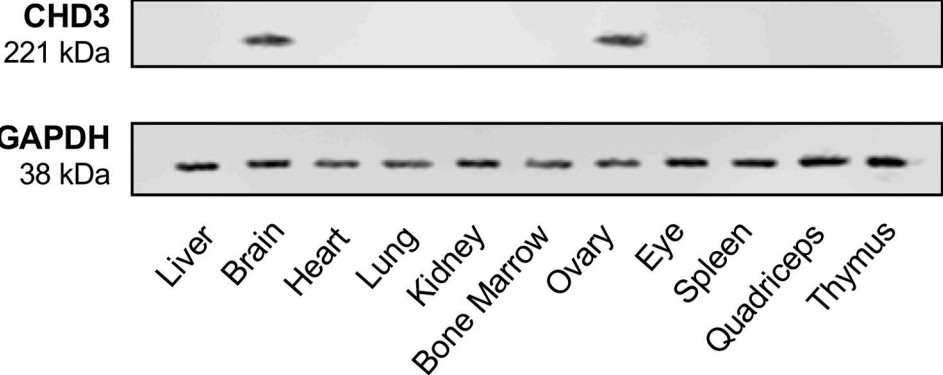

**Fig 4. CHD3 is predominantly expressed in the brain and gonads of adult mice.** Representative immunoblots of tissues from a single male and female wildtype mouse are shown. GAPDH was immunoblotted as a loading control. At least three mice of each gender were analyzed with comparable results.

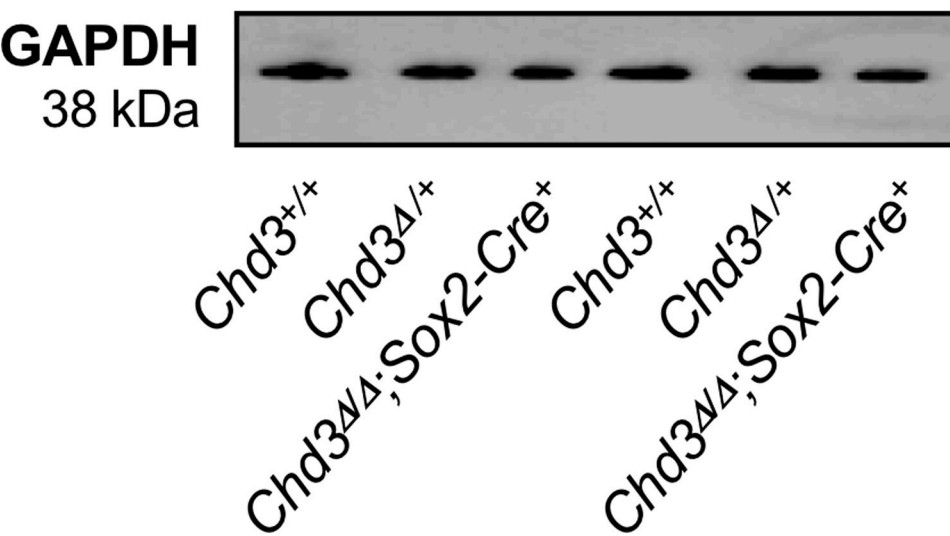

**Fig 5.** ***Chd3* is efficiently excised with the *Sox2-Cre* recombinase.** Representative immunoblots of brains from two sets of wildtype (*Chd3*$^{+/+}$), *Chd3*-heterozygous (*Chd3*$^{\Delta/+}$) and *Chd3*-knockout (*Chd3*$^{\Delta/\Delta}$;*Sox2-Cre*$^{+}$) male mice are shown. Four mice of each genotype were analyzed with comparable results.

## Discussion

This study describes the first report of a conditional allele that facilitates global and cell-specific genetic analyses of CHD3 function in murine tissues. Our data indicate that CHD3 does not play a critical role in embryonic vascular development or adult blood vessel maintenance. This finding stands in contrast to our previous work with CHD4, an alternative NuRD ATPase, which transcriptionally regulates vascular development and integrity at multiple stages of embryonic life [14–17]. At face value, these results indicate that CHD4-NuRD complexes are the predominant or most consequential NuRD complexes in endothelial cells and hematopoietic cells. However, it is possible that CHD3-NuRD complexes play transcriptional regulatory roles in blood vessels under challenge conditions that were not incorporated into this study, such as ischemia, hypertension, inflammation, glucotoxicity or in solid tumors. Future work with constitutive *Chd3*$^{ECKO}$ and inducible *Chd3/4*$^{ECKO}$ mice could address this possibility. It is also interesting to consider the contribution of a third NuRD ATPase—CHD5—toward endothelial cell transcriptional regulation. As we report in this study for CHD3, CHD5 is predominantly expressed in the mouse brain and testes [22–24]. Therefore, CHD5 may also play inconsequential roles in the vasculature. Nevertheless, conditional genetic deletion studies—such as those described in the present study—would be the most rigorous way to assess possible roles for CHD5-NuRD complexes in vascular development.

Even though we found no consequential roles for CHD3 in blood vessels, our study does indicate that global deletion of *Chd3* with a *Sox2-Cre* recombinase line results in partial embryonic lethality. We believe this lethality occurs during embryogenesis because we did not observe death of juvenile mice in our colony prior to weaning. Although we did not perform CHD3 expression analyses on embryonic tissues for this study, our immunoblotting assays indicate that CHD3 is predominantly expressed in the gonads and brains of adult mice, which could indicate a critical role for the protein in gametogenesis or fertility. However, we found *Chd3*-heterozygous mice (*Chd3$^{\Delta/fl}$* and *Chd3$^{\Delta/+}$*) represented at expected numbers at weaning (Tables 4 and 5), indicating that *Chd3$^{\Delta}$* gametes are viable and fertile. Moreover, a recent preprint reports that CHD3 plays no significant role in testis or sperm development when our *Chd3*-flox allele is combined with a primordial germ cell-specific *Ddx4-Cre* [25]. Alternatively, CHD3 is expressed in embryonic mouse brain cortices as early as E15.5 and has been shown through shRNA electroporation studies to influence neuronal migration and laminar identity in late stages of embryonic cortical development [6]. CHD4 likewise influences cortical development, and neuronal deletion of *Chd4* with a *Nestin-Cre* results in lethality associated with microcephaly at birth [6]. Therefore, although the known contributions of CHD3 and CHD4 to cortical development are mechanistically different [6], it is possible that some of our *Chd3*-knockout embryos also die with lethal developmental brain defects.

Altogether, the *Chd3*-flox mice we have generated for this study provide opportunities for future genetic analysis of cell-specific roles for this NuRD-associated chromatin-remodeling enzyme. In particular, temporal neuronal-specific deletion of *Chd3* could further clarify its roles in embryonic and postnatal brain development, which is clinically relevant in light of the recent report that human *CHD3* mutations underlie a neurodevelopmental syndrome characterized by macrocephaly and impaired speech [21]. These and other *Chd3* deletion studies would complement the numerous CHD4 cell-specific studies that have already been reported and would expand our knowledge about roles for NuRD complexes in development and disease.

## Supporting information

**S1 Raw images.**
(PDF)

## Acknowledgments

We thank Tirzah Prince, Robert Pahissa, and Jocelyn Rodriguez for technical help with this project, Katia Georgopoulos (Harvard Medical School) for *Chd4$^{fl/fl}$* mice, and Dr. Roberto Pezza and Griffin lab members for helpful discussions and advice.

## Author Contributions

**Conceptualization:** Courtney T. Griffin.

**Funding acquisition:** Courtney T. Griffin.

**Investigation:** Jun Xie, Siqi Gao, Christopher Schafer, Sarah Colijn, Vijay Muthukumar, Courtney T. Griffin.

**Writing – original draft:** Courtney T. Griffin.

**Writing – review & editing:** Jun Xie, Siqi Gao, Christopher Schafer, Sarah Colijn, Vijay Muthukumar, Courtney T. Griffin.

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
