## [Decision Letter · Decision Letter 0]

27 May 2020

PONE-D-20-12024

The chromatin-remodeling enzyme CHD3 plays a role in embryonic viability but is dispensable for early vascular development

PLOS ONE

Dear Dr. Griffin,

Thank you for submitting your manuscript to PLOS ONE. After careful consideration, I feel that it has merit but does not fully meet PLOS ONE’s publication criteria as it currently stands. Therefore, I invite you to submit a revised version of the manuscript that addresses the points raised during the review process.

Please ensure to include the data requested by the Reviewer, as it is not acceptable to have critical supporting evidence referred to as "data not shown". Also, the speculative statements pointed out by the Reviewer need to either be supported with data or removed from the manuscript.

I will look forward to receiving your revised manuscript.

Kind regards,

Edward E Schmidt

Academic Editor

PLOS ONE

Journal Requirements:

4. We noted in your submission details that a portion of your manuscript may have been presented or published elsewhere:

'The Chd3-flox allele we generated for this manuscript is briefly described in our collaborator's manuscript that is currently under consideration for publication (de Castro et al; Biorxiv; 2019). We do not consider this a dual publication, because generation of the Chd3-flox allele is not described in detail in the de Castro study, and the Cre-recombinase line used in that study is different than the ones used for this study. However, we cite this preprint in our discussion and therefore have uploaded it for reviewer consideration.'

Please clarify whether this publication was peer-reviewed and formally published. If this work was previously peer-reviewed and published, in the cover letter please provide the reason that this work does not constitute dual publication and should be included in the current manuscript.

Reviewers' comments:

Reviewer's Responses to Questions

**Comments to the Author**

1. Is the manuscript technically sound, and do the data support the conclusions?

Reviewer #1: Yes

2. Has the statistical analysis been performed appropriately and rigorously? 

Reviewer #1: Yes

3. Have the authors made all data underlying the findings in their manuscript fully available?

Reviewer #1: Yes

4. Is the manuscript presented in an intelligible fashion and written in standard English?

Reviewer #1: Yes

5. Review Comments to the Author

Reviewer #1: Review of PONE-D-20-12024, “The Chromatin-remodeling enzyme CHD3 plays a role in embryonic viability but is dispensable for early vascular development.”

In this manuscript, J. Xie and colleagues generated a conditional (floxed) allele of murine CHD3, a component of the NuRD chromatin-remodeling complex. Subsequent analysis of Tie2- and Sox2-Cre-mediated deletion of the Chd3-floxed allele revealed no apparent role for CHD3 in embryonic vascular development, as was previously reported for CHD4. Interestingly, a potential requirement for CHD3 activity was suggested by Chd3-knockout mice. Global deletion of CHD3 resulted in partial lethality of mice during gestation, based on observed versus expected Chd3Δ/Δ offspring at weaning. Western blot analyses of tissue-specific expression of CHD3 revealed that CHD3 is primarily expressed in adult brain and gonads. Based on these findings, the authors conclude that CHD3 may play a role in embryonic neurodevelopment.

Comments:

1. Although the authors showed that efficient, Cre-mediated deletion of the Chd3-flox allele and lack of CHD3 protein in adult Chd3Δ/Δ brain tissue, validation of the Chd3-flox allele would be augmented by addition of the Southern blot analysis of targeted ES cells and PCR-based analyses of Cre-mediated recombination. The authors list these analyses as “data not shown.” In my opinion, this data should be included in Figure 1.

2. In the discussion section, the authors suggest that “temporal neuronal-specific deletion of Chd3 could further clarify its roles in embryonic and postnatal brain development…,” although no evidence was presented in the manuscript that CHD3 is actually expressed in developing neuronal tissue. Evidence of embryonic, tissue- and developmental stage-specific CHD3 expression would be useful in guiding future studies using Chd3-flox mice, and should be included in this manuscript.

6. PLOS authors have the option to publish the peer review history of their article (what does this mean?). If published, this will include your full peer review and any attached files.

Reviewer #1: No

---

## [Author Response · Author response to Decision Letter 0]

15 Jun 2020

Editorial/Journal Requests:

1. Ensure that the manuscript meets PLOS ONE’s style requirements:

We have read the requirements carefully and have reformatted the manuscript and figures accordingly.

2. “Data not shown” cannot be used in the manuscript:

The phrase “data not shown” was originally located at the end of our first “results” paragraph and referred to Southern blotting of ES cells to validate homologous recombination of the Chd3-flox targeting vector prior to blastocyst injection. This Southern blotting was performed by Cyagen Biosciences, with whom we contracted to make our Chd3-flox mice. Cyagen never sent us the Southern blot images, so we cannot include them and have therefore removed the phrase “data not shown” from the manuscript and indicated that Southern blotting was performed by Cyagen. In the same sentence, “Data not shown” also referred to PCR genotyping validation of Chd3-excision events in our mice carrying a Cre recombinase. We have now included these genotyping data in Fig 1E.

3. Show all original uncropped and unadjusted images underlying all blot or gel results:

We have now provided all original blots and gels as Supporting Information. Please note that in collecting our original blots for Figure 4B (female mouse tissues blotted for CHD3 and GAPDH), we were unhappy with the aesthetics of the background bands on the GAPDH blot. Therefore, we re-ran the same lysates and re-probed them for CHD3 and GAPDH. The new data recapitulate the original data very closely but give a better exposure for CHD3 and a cleaner blot for GAPDH. Therefore, we have substituted these new blots for the original ones that we had submitted for Figure 4B (see uploaded "Response to Reviewer" document, which includes a figure showing the side-by-side original and updated blots).

 

4. Dual publication issue:

As we reported upon submission, the Chd3-flox allele we generated for this manuscript is briefly described in our collaborator’s manuscript that was reviewed by Development in April. Our collaborator (Dr. Pezza) is still revising the manuscript for resubmission to Development. Since it (1) has not yet been re-reviewed and accepted, (2) does not describe the generation of the Chd3-flox allele in detail, and (3) does not use the same Cre-recombinase lines used in this study, we do not consider this a dual publication. However, we have cited the preprint of that study in our discussion (de Castro et al; bioRxiv; 2019).

Reviewer Comments:

1. “Although the authors showed that efficient, Cre-mediated deletion of the Chd3-flox allele and lack of CHD3 protein in adult Chd3��/� brain tissue, validation of the Chd3-flox allele would be augmented by addition of the Southern blot analysis of targeted ES cells and PCR-based analyses of Cre-mediated recombination. The authors list these as “data not shown.” In my opinion, this data should be included in Figure 1.”

We appreciated this suggestion and refer the Reviewer to our response to the Editor (see #2 under “Editor/Journal Requests” above and new Fig 1E).

2. “In the discussion section, the authors suggest that “temporal neuronal-specific deletion of Chd3 could further clarify its roles in embryonic and postnatal brain development…,” although no evidence was presented in the manuscript that CHD3 is actually expressed in developing neuronal tissue. Evidence of embryonic, tissue- and developmental stage-specific CHD3 expression would be useful in guiding future studies using Chd3-flox mice, and should be included in this manuscript.”

In response to this request, we have revised our Discussion (2nd to last paragraph) to cite a published report that CHD3 is expressed in embryonic mouse brain cortices as early as E15.5 (Nitarska et al, 2016, Cell Reports, 17, 1683-1698). This study also shows that electroporation of CHD3 shRNA into developing cortices compromises neuronal migration and laminar identity during development. We feel that this important citation fulfills the Reviewer’s request in providing evidence for CHD3 expression in developing neuronal tissue and in demonstrating that CHD3 is functionally important for embryonic brain development.

---

## [Editor Report · Decision Letter 1]

23 Jun 2020

The chromatin-remodeling enzyme CHD3 plays a role in embryonic viability but is dispensable for early vascular development

PONE-D-20-12024R1

Dear Dr. Griffin,

I am pleased to inform you that your manuscript has been judged scientifically suitable for publication and will be formally accepted for publication once it meets all outstanding technical requirements.

Kind regards,

Edward E Schmidt

Academic Editor

PLOS ONE
---

## [Editor Report · Acceptance letter]

26 Jun 2020

PONE-D-20-12024R1 

The chromatin-remodeling enzyme CHD3 plays a role in embryonic viability but is dispensable for early vascular development 

Dear Dr. Griffin:

I'm pleased to inform you that your manuscript has been deemed suitable for publication in PLOS ONE. Congratulations! Your manuscript is now with our production department. 

Kind regards, 

on behalf of

Dr. Edward E Schmidt 

Academic Editor

PLOS ONE